# Investigating the Crime Scene—Molecular Signatures in Inflammatory Bowel Disease

**DOI:** 10.3390/ijms241311217

**Published:** 2023-07-07

**Authors:** Vibeke Andersen, Tue B. Bennike, Corinna Bang, John D. Rioux, Isabelle Hébert-Milette, Toshiro Sato, Axel K. Hansen, Ole H. Nielsen

**Affiliations:** 1Molecular Diagnostic and Clinical Research Unit, University Hospital of Southern Denmark, Institute of Regional Research, University of Southern Denmark, 5000 Odense, Denmark; tbe@hst.aau.dk; 2Institute of Molecular Medicine, University of Southern Denmark, 5000 Odense, Denmark; 3Medical Microbiology and Immunology, Department of Health Science and Technology, Aalborg University, 9000 Aalborg, Denmark; 4Institute for Clinical Molecular Biology, Christian-Albrecht’s University, 24105 Kiel, Germany; c.bang@ikmb.uni-kiel.de; 5Department of Medicine, Université de Montréal, Montreal, QC H3C 3J7, Canada; john.rioux@inflammgen.org (J.D.R.); isabelle.hebert-milette@inflammgen.org (I.H.-M.); 6Montreal Heart Institute Research Institute, Montreal, QC H1T 1C8, Canada; 7Department of Gastroenterology, Keio University School of Medicine, Tokyo 160-8582, Japan; t.sato@keio.jp; 8Experimental Animal Models, Department of Veterinary and Animal Sciences, University of Copenhagen, 1870 Frederiksberg, Denmark; akh@sund.ku.dk; 9Department of Gastroenterology, Herlev Hospital, University of Copenhagen, 2730 Herlev, Denmark

**Keywords:** biomarkers, Crohn’s disease, gut microbiota, intestinal barrier, personalised medicine, ulcerative colitis

## Abstract

Inflammatory bowel diseases (IBD) are without cure and troublesome to manage because of the considerable diversity between patients and the lack of reliable biomarkers. Several studies have demonstrated that diet, gut microbiota, genetics and other patient factors are essential for disease occurrence and progression. Understanding the link between these factors is crucial for identifying molecular signatures that identify biomarkers to advance the management of IBD. Recent technological breakthroughs and data integration have fuelled the intensity of this research. This research demonstrates that the effect of diet depends on patient factors and gut microbial activity. It also identifies a range of potential biomarkers for IBD management, including mucosa-derived cytokines, gasdermins and neutrophil extracellular traps, all of which need further evaluation before clinical translation. This review provides an update on cutting-edge research in IBD that aims to improve disease management and patient quality of life.

## 1. Introduction

Inflammatory bowel disease (IBD) is a chronic inflammatory disorder of the gastrointestinal tract and includes ulcerative colitis (UC) and Crohn’s disease (CD) [1,2,3,4]. The incidence of IBD is rising worldwide, particularly in Asia, and prevalence is predicted to reach 1% by 2030 in many regions [5,6]. The etiology remains unknown.

Managing IBD is challenging mainly due to extreme heterogenicity in the disease trajectories between patients, which affects diagnosis, optimal treatment choice and prediction of disease course and complications [7,8]. Establishing the correct diagnosis of an IBD can be difficult and may delay establishing an optimal treatment course associated with adverse disease outcomes [9,10]. Despite recent advancements in targeted treatment options, remission rates are as low as 20–30% in some studies [11]. One in three IBD patients will require surgery within five years of diagnosis due to accumulating tissue damage resulting from insufficient disease control [12,13]. Consequently, for many patients, the treatment and management of IBD is suboptimal, resulting in a marked decrease in quality of life for patients and their families and ever-increasing costs to society due to loss of income, work absenteeism and healthcare expenses [14,15].

Personalised medicine aims to address the diversity of IBD by tailoring treatment strategies to individual patients based on important factors involved in the disease mechanisms. Since 2019, cutting-edge research has demonstrated that the effects of diet depend on patient factors and gut microbial activity. Furthermore, a range of biomarker candidates have been identified that need evaluation for clinical translational potential [16,17,18]. This review synthesises the newest published research and provides the latest understanding of the relationship between diet, gut microbes and the immune system of the gut barrier (Box 1). Ultimately, this research aims to improve disease management and the patient’s quality of life. 

Box 1Research in context and outstanding questions.What was known before:Diet, gut microbes and patient immune factors are essential factors for IBD initiation and progressionNearly every second case of IBD can be prevented by a healthy lifestyle including a healthy dietThe enormous diverse nature of patients poses challenges in disease managementWhat this review adds:An update on cutting-edge research identifying molecular profiles to reflect patient diversity based on interactions between diet, gut microbes and the patient’s immune system (Figure 1)Patient diet interferes with gut inflammation depending on the patient immune status, genetics and gut microbiomeCertain faecal metabolome profiles, specifically short-chain fatty acids, are better indicators of IBD phenotypes compared to faecal metagenome or metatranscriptome.Potential biomarkers derived from the gut mucosa (e.g., cytokines, gasdermins, neutrophil extracellular traps, Faecalibacterium prausnitzii) and circulating biomarkers (e.g., redox status, N6-methyladenosine modification) are discussedOutstanding research areas:Better understanding of specific microbiota–patient interactions by characterising mucosa-associated microbiota and the accompanying immune responsesBetter molecular characterising of IBD sub-phenotypes such as patients with specific disease courses, complications and other immune-related diseasesBetter molecular characterisation of specific IBD phenotypes improving from specific dietary and drug interventionsBetter molecular understanding of patient diversity based on careful phenotypic patient stratificationCombining specific clinical information for IBD phenotypes with omics data using data integration as a way forward to identify clinically useful biomarkersProspective longitudinal observational studies of biomarkers considered for clinical translation to validate potential biomarkersReplication and validation of promising biomarkers in patient cohorts from different geographic regions (e.g., Asia) ultimately leading to clinical translation

## 2. Identifying Biomarkers Based on Diet–Gut Microbes–Patient Interactions

Biomarkers with documented diagnostic and prognostic value have the potential to optimise disease management through personalised medicine strategies. Such biomarkers reflect the diversity among patients and identify groups of patients constituting a distinct phenotype characterised by specific properties and treatment needs. Personalised medicine may, in this way, improve diagnostic accuracy, track disease course, predict complications, and enable highly personalised and targeted treatment strategies (Figure 1). With a range of new drugs available and in the pipeline, tools are urgently needed to select the most effective treatment for the individual patient [11,19,20]. In addition, biomarker-based dietary intervention could offer an additional non-immunosuppressive treatment option for patients [21,22]. Consequently, a currently unmet need is to identify, evaluate and implement promising IBD biomarkers in personalised medicine [16,17,18].

Although accumulating evidence points to interactions among diet, gut microbes and immune factors on the gut barrier as key elements controlling IBD, many studies have only individually characterised the environmental, gut microbial and patient factors, with interactions between these factors remaining unclear [23]. Given the multifactorial biology of IBD, a new and promising approach involves identifying biomarkers that reflect these interactions. Indeed, new and emerging high-throughput technologies and data integration methods can create detailed biological omics datasets and combine them with clinical and lifestyle information. Furthermore, these techniques may characterise specific IBD phenotypes and identify potential biomarkers and new drug targets (Figure 1) [7,23,24,25]. For example, microbiome risk profiles may have the potential to identify similar groups of patients [24].

## 3. Essential Factors Involved in IBD

The gut epithelial barrier separates the luminal contents from the underlying tissue layers and immune cells. It controls the interactions between the patient’s immune system, the gut microbiota and environmental factors such as food components and is implicated in IBD [26]. This review argues that biomarkers reflecting these interactions (lifestyle, gut microbes and the gut barrier immune system) can serve as diagnostic, prognostic, predictive or monitoring properties to enhance the management of IBD (Figure 2). Table 1 and Table 2 show examples of the key factors involved in patient–diet–microbial interactions. 

### 3.1. Diet and IBD

Characterising the impact of diet on any disease is challenging due to the complex composition of modern diets and unreliability of self-reporting. In addition, an in-depth understanding of the functional effect of diet is lacking [62,63]. Nevertheless, strong associations between diet and IBD have been demonstrated, and recent studies have highlighted that lifestyle including diet strongly influences IBD risk [64,65,66]. In contrast, few randomised clinical trials have been performed investigating the effect of diet on patients with established IBD [31,67,68]. Prospective studies have demonstrated that a Western diet, characterised by a high intake of animal-based foods, processed foods, food additives, alcohol and sugar, is associated with IBD and increases the occurrence of flare-ups compared to a healthy diet [29,30]. High intake of ultra-processed foods has also been associated with an increased risk of IBD [28]. On the other hand, a plant-based diet such as a Mediterranean diet was reported to reduce inflammation in IBD [31]. However, other studies have failed to demonstrate an association between a specific diet and IBD, and the association remains somewhat unclear [69,70]. Consequently, evidence-based nutritional recommendations for the individual patient are scarce [62,63].

### 3.2. The Gut Microbiota and IBD

Compared with healthy control individuals, patients with IBD consistently demonstrate gut microbiota alterations. The changes include dysbiosis, characterised by lowered bacterial α-diversity (i.e., fewer defined microbial species) and altered β-diversity (i.e., significant changes in microbial species composition) [71]. The systematically described collection of microbes is known as the microbiota, while the term microbiome includes their pool of functional genes. The loss of resident microbial species, termed the “disappearing microbes”, might help to explain the rising incidence of chronic diseases in industrialised countries [24].

Many studies have found that patients with IBD have an increased abundance of *Escherichia coli* and *Fusobacterium* spp. known to promote inflammation by the adhesion and invasion of the colon epithelium. Further, a lowered abundance of the short-chain fatty acids (SCFA) producers *Faecalibacterium prausnitzii* and *Akkermansia muciniphila* has also been observed [72,73]. Reports indicate that a high abundance of the class Actinobacteria and the associated genus *Bifidobacterium* are protective against UC [74]. In contrast, species such as *Ruminococcus gnavus* and *R. torques* typically increase gut inflammation through their production of a TNF-α inducing polysaccharide and are abundant in patients with IBD [75]. 

However, apart from lower diversity, studies report inconsistent patterns of gut microbiota alterations in IBD [24]. This inconsistency is, at least in part, due to the heterogeneity of the disease [34]. Additionally, numerous factors affect human gut microbiota composition, including the sampling method, geographic location and patient factors, such as genetics, sex, age, diet, stool consistency and other lifestyle factors [24,76]. 

Importantly, changes observed in the gut microbiota can be a consequence or a cause of IBD. Recent data support the key role of a specific bacterium, *Klebsiella pneumonia*, in IBD [40]. It was found in approximately 40% of patients, the abundance correlated with disease activity, and its transfer resulted in colitis in an animal model [40]. While the exact role and mechanisms remain unclear, it is conceivable that *K. pneumonia* may be involved in the etiology of a subset of IBD. Complicating the aspect of causality further is that there seems to be a critical window in early life in which perturbation of the microbiome has a substantial effect on disease development [77].

Bacterial members of the microbiota are not the only microorganisms that can be altered in IBD [78,79,80]. Fungi, archaea and viruses can also significantly affect the gut immune response to IBD, although they only account for a minor proportion of the mammalian gut microbiota [78]. In particular, the faecal mycobiome differed between patients with CD and UC and between patients experiencing a flare compared to those in remission, where the mycobiome more closely resembles a healthy mycobiome [80]. In addition, viruses have been associated with IBD by activating the immune system following invasion and replication within the epithelial cells [81]. Similarly, phages can indirectly affect immune cells and other cell types through infected bacteria [40]. However, methodological biases may still complicate interpretation. 

### 3.3. Gut Epithelium Barrier and Immune System in IBD

The gut epithelial barrier controls the interaction between the gut microbiota and food components on the one hand and the patient immune system on the other (Figure 2) [27]. In IBD, this barrier is compromised, giving rise to the condition commonly described as a “leaky gut”. The leaky gut is probably a key pathological factor in IBD as it has been found to precede diagnosis [61].

The fundamental structures of the gut epithelial barrier are, from the luminal side, the mucus layer and the intestinal epithelial cells lining. The colonic mucus is a two-layered gel-like structure produced by goblet cells comprising highly glycosylated mucin proteins. In the healthy gut, commensal microorganisms interact with the outer mucus layer and cannot reach the inner mucus layer or epithelial cells [82]. Functional mucus glycosylation is essential for feeding microbes, and altered glycosylation patterns contribute to pronounced alterations in the gut microbiota [83]. In IBD patients, altered spatial patterns have been found to contribute to microbiota dysbiosis [82]. Thus, it is becoming increasingly evident that microbiota–host interactions depend highly on the microbial communities’ nature and spatial organisation [84]. Nevertheless, few studies have analysed the luminal or mucosa-associated microbiota, which are in close contact with the gut immune system and differs from the stool microbiota [85]. The gut epithelium consists of cells capable of activating the immune system when in contact with dietary materials, microbial components or metabolites [1]. For example, pattern recognition receptors and G protein-coupled receptors on intestinal epithelial cells respond to specific microbial structures and metabolites [86,87,88,89]. In recent years, new epithelial cell types such as intercrypt goblet cells [90], microfold-like (M-like) cells [91], BEST4+ cells [92] and Tuft cells have been identified. Tuft cells appear to be critical for specific immunologic responses [93,94]. M-like cells are rarely found in healthy colons but are reported to be expanded 17-fold in inflamed colons [91]. BEST4+ cells were identified as a new population of human intestinal epithelial cells by single-cell RNA-seq technology. Histologic analysis revealed their localization in the crypt top. The functional role of BEST4+ cells remains unknown, but they may be associated with bicarbonate export and a pH-sensing function based on their gene expression. Finally, the gut epithelial basement membrane is a specialized matrix that supports and separates the epithelial cells from the interstitial space and is also considered important in maintaining the epithelial barrier [2]. Understanding the host–microbial interactions at this surface will likely prove critical to gain deeper biological insights into the etiology of IBD and identifying clinically useful biomarkers. 

In addition, understanding the role of the gut microbiome in the brain, joints and liver is emerging, indicating that the microbiota is a driving factor for altered cell trafficking, a crucial step for the onset and progression of extraintestinal conditions in IBD [95]. 

## 4. Interactions between Diet, Gut Microbiota and Host Factors

Diet can affect the patient immune system either directly or indirectly by changing the microbial activity, and patient factors can impact the microbial function and the effects of diet. Similarly, gut microbes can affect the immune system directly through contact with the epithelium or indirectly through the production of various molecules subsequently absorbed by the host [1]. 

### 4.1. Linking Diet with Gut Microbiota and Patient Factors 

As mentioned, diet strongly affects gut microbial function and is associated with inflammation. IBD intervention studies have demonstrated that a Mediterranean-based or low-fat diet resulted in a healthier microbial composition (i.e., *F. prausnitzii* enrichment) [31,32,67]. Moreover, in conditions other than IBD, the Mediterranean diet correlated with inflammation suppression, increased abundance of *F. prausnitzii* and *Roseburia* and decreased abundance of *R. gnavus*, *Collinsella aerofaciens* and *R. torques* [96,97].

Dietary studies are complex and may be further complicated by the finding that gut microbiota composition impacts the effects of diets. For example, a recent study found that the protective effect of a Mediterranean diet on cardiometabolic risk was higher in participants lacking specific critical microbes (*Prevotella copri*) [98]. 

Combining diet and microbiota transplantation may prove successful. Consequently, randomised clinical trials are underway combining diet and microbiota transplantation in patients with UC [99,100]. One study reported that combining an anti-inflammatory diet and weekly faecal microbiota transplantation for eight weeks was superior to medical therapy [100]. Therefore, studies on the impact of diet on IBD should consider the gut microbiota composition at baseline.

The concept was further developed in a study demonstrating that both gut microbial activity and patient status impacted the effects of diet. In IBD, increasing dietary fibre intake could be beneficial for certain patients despite not being recommended for patients with symptoms of stenosis due to the risk of needing surgical intervention [101]. However, dietary fibres are heterogenous compounds, and different fibre types can evoke different biologic responses [68,102]. A wholegrain fibre diet increased the level of butyrate in overweight individuals [103]. Accordingly, a recent study found that a dietary fibre’s effect depended on the fibre type, the patient immune status and the fermentative capacity of their gut microbiota [21]. Armstrong et al. found that certain β-fructans such as fructo-oligosaccharide and inulin, but not barley, maltodextrin, or starch, triggered a pro-inflammatory response in peripheral blood mononuclear cells from healthy donors as evidenced by the increased release of IL-1β. The authors cultured colonic biopsies from paediatric patients with IBD with both an active and a quiescent disease and from control subjects without IBD. Culturing in the presence of fructo-oligosaccharides significantly increased IL-1β secretion in colonic biopsies from patients with active IBD and, to a lesser extent, from those with a quiescent disease but decreased IL-1β secretion in biopsies from control subjects without IBD [21]. 

Consequently, future dietary recommendations might be tailored to an individual’s immune and gut microbial function profiles. Moreover, colonic IBD might be more amenable to dietary interventions than CD localised in the small intestine because of the diet’s interaction with the microbial composition and formation of microbial metabolites at the disease site [104]. 

### 4.2. Gut Microbiome Can Affect the Patient Immune System

Specific microbial profiles or species have been suggested as biomarkers in IBD, as some have been associated with IBD activity or treatment response (e.g., Faecalibacterium and Bifidobacteria) or nonresponse (e.g., Veillonella and Fusobacterium) [24,34] (reviewed in [38]) (Table 1 and Appendix A). For example, a study of patients with IBD treated with tumour necrosis factor (TNF) inhibitors reported lower abundances of SCFA producers (particularly of the class Clostridia) and higher abundances of pro-inflammatory bacteria and fungi (e.g., genus Candida) among non-responders than among responders [33]. In addition, another study of patients with IBD treated with anti-cytokine (anti-TNF or anti-IL-12/23) or anti-integrin drugs used a multi-omics analysis of stools to identify associations with drug responses after 14 weeks [39]. The authors found that baseline microbial richness was associated with the degree of response to anti-cytokine therapy, and responders had a greater abundance of butyrate-producing microbial species in the colon [39]. Unfortunately, baseline multi-omic profiles were only available for a few participants, illustrating that relatively small sample sizes are a major limitation of these studies, along with significant heterogeneity, preventing robust validation [39].

Interestingly, it has been found that the faecal metabolome was better at identifying IBD features compared to the faecal metagenome, faecal metatranscriptome or the faecal proteome and could even discriminate between UC, CD, ileal and colonic inflammation [24]. The reason is thought to be the complexity of the microbiome, where several metabolically active microorganisms work together in complex microbial communities to ferment the contents of the gut lumen after digestion [84]. Together, these organisms contribute to the ecosystem where microbes exchange or compete for nutrients, signalling molecules, or immune-evasion mechanisms through complicated and often unclarified interactions [105]. 

Consequently, microbial metabolites may quantify the diversity among patients. In particular, SCFAs such as butyrate have been investigated as potential biomarkers [25,106]. Generally, in IBD, lower SCFA levels and fewer SCFA-producing bacteria are measured in faeces compared to healthy control subjects. [106] However, results are inconsistent, and in children with CD, remission was not associated with increased SCFAs despite observing an increase in SCFA synthesis pathways [107]. Currently, the potential role of SCFAs as a biomarker in IBD is unclear, but research has shown that SCFAs can also affect tissues and organs beyond the gut through systemic circulation and affect future generations through epigenetic imprinting in utero [108].

### 4.3. Patient Factors Affecting the Gut Microbial Function

IBD-associated genes are involved in the interaction between the microbiota and the mucus layer and may disrupt key intracellular processes, including bacterial handling (Appendix A) [109,110,111,112,113,114]. Some examples of IBD-associated genes that affect cell apoptosis and apical junction function, which are essential for the integrity of the epithelial barrier, include C1orf106, RNF186, DUSP16 (polygenic IBD) and ALPI, GUCY2C and TTC7A (monogenic IBD) [112,113,115,116]. In addition, altered barrier functions contribute to dysregulated intestinal epithelial homeostasis in IBD.

Moreover, the intestinal epithelium regulates the microbial environment through the secretion of antimicrobial peptides, such as lysozyme from Paneth cells. However, IBD-associated genes, including NOD2, ATG16L1 and ALPI, impair this process and change the gut microbiota composition in patients with IBD [115]. Consequently, genetics may impact gut inflammation through changing the gut microbiota [117]. This conclusion was supported by a twin study that utilizes the fact that healthy twins with a co-twin with established IBD have increased risk of IBD compared with the general population. The study found that the gut microbiota composition of the healthy twin was closer to that of IBD patients than healthy control individuals including *F. prausnitzii* and butyrate biosynthesis pathways [118]. Another study of genetics and microbiome composition in patients and controls from families with IBD further supported that genetics impacts gut microbial composition. The linkage study found that distinct chromosomal regions are linked to different microbiome traits in IBD families [119]. 

Another interesting example is the *FUT2* gene [120]. Approximately 20% of individuals of European ancestry carry the IBD-associated risk variant of *FUT2* (FUT2 non-secretors). FUT2 non-secretors lack terminal fucose residues in their gut mucin, on which mucus-degrading bacteria feed, resulting in decreased stool microbiome diversity compared to FUT2 secretors [120]. Individuals with the *FUT2* risk gene demonstrate low mucosa-associated abundance of butyrate-producing bacteria *F. prausnitzii* and low microbiota diversity. 

The intestinal epithelium can also accumulate somatic mutations during chronic inflammation that affect epithelial function [44,45,46]. During chronic inflammation, the intestinal epithelium is exposed to proinflammatory cytokines. A recent genetic analysis of colonic epithelium tissue from patients with UC revealed an accumulation of somatic mutations. Interestingly, these somatic mutations were associated with interleukin (IL)-17 signalling pathway components, including PIGR, NFKBIZ, IL17RA and TRAF3IP2 [44,45,46]. The expansion of mutant clones was typically observed in patients with UC who developed UC-associated cancers, possibly reflecting long-term exposure to chronic inflammation. Although counterintuitive, these mutations were found exclusively in the nontumour epithelium, suggesting a tumour-suppressive function [45]. These findings demonstrate that intestinal epithelium can accumulate somatic mutations, potentially affecting intestinal epithelium function. 

Another way to affect gene function is RNA metabolism, such as RNA transcription. Whereas correct RNA splicing is a prerequisite for correct gene transcription and protein function, extensive deregulation of splicing precision has been found in UC [48]. Thus, the level of heterochromatin protein 1γ (HP1γ), a regulator of gut inflammatory genes in response to enterobacteria, was low in UC, leading to improper RNA splicing (high splicing noise) [48]. Further, high splicing noise in the gut correlated with disease activity measured by a histological severity score and mucosal healing after treatment [48]. However, whereas epigenetics and RNA modifications offer a way that the environment can affect gut inflammation and thus are attractive for further exploration as biomarkers, their role in IBD is complicated, and their clinical potential is not clarified.

Oxidative stress and disrupted redox signalling connect the epithelial barrier function, mucus production and the gut microbiome [52]. IBD is characterised by an inability to cope with the increased oxygen production related to gut inflammation leading to redox imbalance. In addition, accumulating evidence has linked the hypoxia-inducible factor (HIF)-1α pathway to compromised mucus production and gut epithelium function, ultimately leading to tissue damage [52]. Generally, patients with IBD exhibit more facultative anaerobes and fewer obligate anaerobes, causing tremendous alterations of fermentation processes and disruptions of microbial transcription [73]. These observations demonstrate that patient factors interact with gut microbial function in relation to IBD. 

### 4.4. Regulation of Mucosal Factors 

Epigenetics refer to the reversible and dynamic changes to the genome that does not involve alteration of the nucleotide sequence. Environmental factors can result in epigenetic modifications that regulate gene expression and affect IBD [121]. Epigenetic modifications, including mRNA modifications, contribute to the regulation of gene expression by influencing mRNA transcription and processing, stability, translation and localization. N6-Methyladenosine (m6A) methylation is the most common, well-understood mRNA modification in the patient–gut microbiota crosstalk [122]. Recently, it was found to play a key role in the development and progression of IBD [47]. Specifically, the m6A modification affects the microbiota by regulating intestinal mucosal immunity and barrier function, as well as intestinal epithelial cell apoptosis and autophagy. However, animal studies have demonstrated that the enteric microbiome can also mediate m6A modification [47]. The gut microbiota can thereby induce epigenetic alterations in the host. Given the role of m6A methylation in IBD, it has been suggested as a predictive biomarker in IBD, but its potential has not yet been evaluated [47].

### 4.5. Mucosal Biomarkers Reflecting Diet–Gut Microbes–Patient Interactions

Although further research to clarify the function is needed, potential biomarkers reflecting diet–microbe–immune system interactions in IBD have been suggested (Table 1). Faecal calprotectin is an established clinical biomarker for distinguishing inflammatory and non-inflammatory gut conditions and tracking IBD disease activity [49,123]. Calprotectin consists of two subunits, S100A8 and S100A9, derived from neutrophils. The expression is strongly increased by exposure to bacterial antigens, such as lipopolysaccharides, lipoprotein and inflammatory mediators, fuelling inflammation by activating the Toll-like receptor-4 and other molecules. Calprotectin thereby orchestrates an inflammatory response at the mucosal surface [49]. Two other potential biomarkers, lactoferrin and myeloperoxidase, are also derived from neutrophils and have anti-microbial activity, e.g., inducing phagocytosis [58,59].

Calprotectin is also a component of neutrophil extracellular traps (NETs) alongside antimicrobial neutrophil granules, cytoplasmic proteins, and DNA filaments. NETs are essential for the inflammatory cascade as they are formed during the first steps of the innate immune response, initiating the general immune response. NETs are produced by neutrophils in the colonic mucosa after contact with microbes or microbial products and they trap and eradicate extracellular bacteria and fungi [26,56,124]. NETs have recently been associated with UC and have been found to sustain inflammation [125,126], NETosis, the process through which neutrophils extrude NETs, is mediated by a gasdermin, gasdermin D-amino-terminal [53]. Gasdermins are a family of structurally related proteins that can modify interactions with gut bacteria by inducing pyroptosis of cells infected with intracellular bacteria [54]. Gasdermin-dependent pyroptosis of intestinal epithelial cells is pathogenic in IBD, causing loss of mucosal integrity (by killing epithelial cells) and mediating the release of inflammatory mediators [53]. Interestingly, IBD-associated gasdermin B gene variants confer functional defects by disrupting epithelial repair, establishing gasdermin B as a critical factor for restoring epithelial barrier function and resolving inflammation [53]. Although calprotectin, NETs, and gasdermins are promising biomarkers due to their role in activating the innate immune system and initiating inflammation, the biological functions are far from clear, and their role as biomarkers remains to be clarified [57,127,128]. 

Finally, new potential biomarkers have been proposed mainly for measuring disease activity (Table 2) [51,58,129,130]. For example, leucine-rich alpha-2 glycoprotein can assess endoscopic activity in CD patients and is a reliable marker of endoscopic remission [51]. Another study suggested that stool chymotrypsin C, gelsolin and rho GDP-dissociation inhibitor 2 (RhoGDI2) correlate with the levels of intestinal inflammation [130]. The authors found that gelsolin and rhoGDI2 in CD, and rhoG in UC, had higher sensitivity and specificity than faecal calprotectin in discriminating between patients and controls. However, the extent that these biomarkers can contribute to management of IBD remains unclear. 

It is clear that combining clinical and omics data can improve diagnostic and prognostic accuracy. For example, combining faecal calprotectin levels with a metagenomic profile into a predictive model improves the prediction of response to therapy and risk of pouchitis in patients with IBD [24]. 

Although some factors have already been suggested (Table 1), a clearer understanding of interactions between diet, gut microbes, intestinal barrier, and the immune system may allow a more thorough clinical evaluation. 

## 5. Conclusions and Future Directions

Despite impressive progress in developing novel biologics and small molecules to treat IBD, it is increasingly apparent that preventing bowel damage and disease complications remains a clinical challenge. Specifically, there is a lack of validated biomarkers in all major IBD areas. 

Therefore, this review synthesises state-of-the-art research addressing the unmet needs for rational management of IBD. Thus, biomarkers reflecting interactions between diet, gut microbes and the patient immune system, essential factors for IBD initiation and progression, are considered successful strategies to advance personalised medicine in IBD. Consequently, research characterising these interactions has been increasingly prioritised, fuelled by advances in technology and data integration.

Cutting-edge research has highlighted the link between diet, gut microbiome and patient immune status in relation to gut inflammation. First of all, a vast range of potential biomarkers for managing IBD has been identified (Table 1). Further, generally, the faecal metabolome, such as SCFA, better reflects IBD phenotypes than the faecal metagenome or metatranscriptome. Combining various levels of information, such as clinical information and omics data, advance the identification of accurate biomarkers. In particular, investigating gut mucosa (the crime scene) contribute to our understanding of the diet-microbiome-patient immune system interactions and identification of clinically useful biomarkers. Presently, work has to be conducted to replicate, validate and translate promising biomarkers to clinical use. Recommendations for future research in the area are outlined in Box 1. For example, future strategies for better management of IBD should include characterising diversity among patients by, e.g., analysing specific genetic variants related to the mucosal barrier and microbial handling together with microbial risk scores indicative of inflammation. In addition, the changing patterns of IBD across the world such as the rising prevalence of IBD in Asia can provide insights into IBD causes [131].

Revealing diet–gut–microbe patient interactions is crucial for selecting clinical biomarkers that reflect the diversity of patients with IBD and depict specific IBD phenotypes. Thus, biomarkers can assist in diagnosis or tailoring disease interventions for an individual patient so that patients in the future can anticipate earlier and more precise diagnoses, prognoses and individualised treatment strategies, including dietary and pharmacologic interventions. In conclusion, biomarkers based on diet–gut microbiota–patient immune system interactions offer enormous potential for patients with IBD and the healthcare system as a whole. 

## 6. Search Strategy and Selection Criteria

Searches of PubMed identified data for this review using the search terms: Crohn’s disease, inflammatory bowel disease, ulcerative colitis, personalised medicine, biomarker, diet, nutrition, microbiome, microbiota, omics, proteomics, transcriptomics, metabolomics, faeces, stool, blood, plasma, serum, mucosa, pathophysiology alone or combined. Only articles published in English after 2019 were included. In addition, backward citation searching was completed on included references.

## Figures and Tables

**Figure 1 ijms-24-11217-f001:**
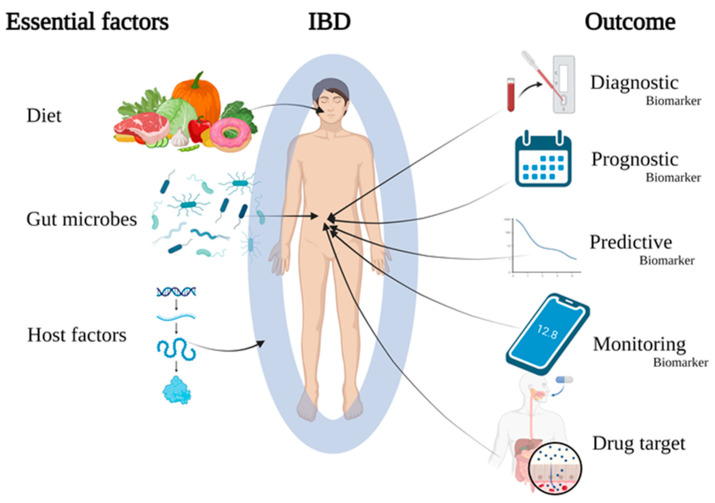
Graphical abstract demonstrating how diet, gut microbes and patient factors affect individuals at risk of IBD or diagnosed with IBD. Understanding the link between these factors is crucial to identify molecular signatures, create diagnostic, prognostic, predictive and disease-monitoring biomarkers, and develop new drugs to manage IBD. Created with Biorender.com (accessed on 2 May 2023).

**Figure 2 ijms-24-11217-f002:**
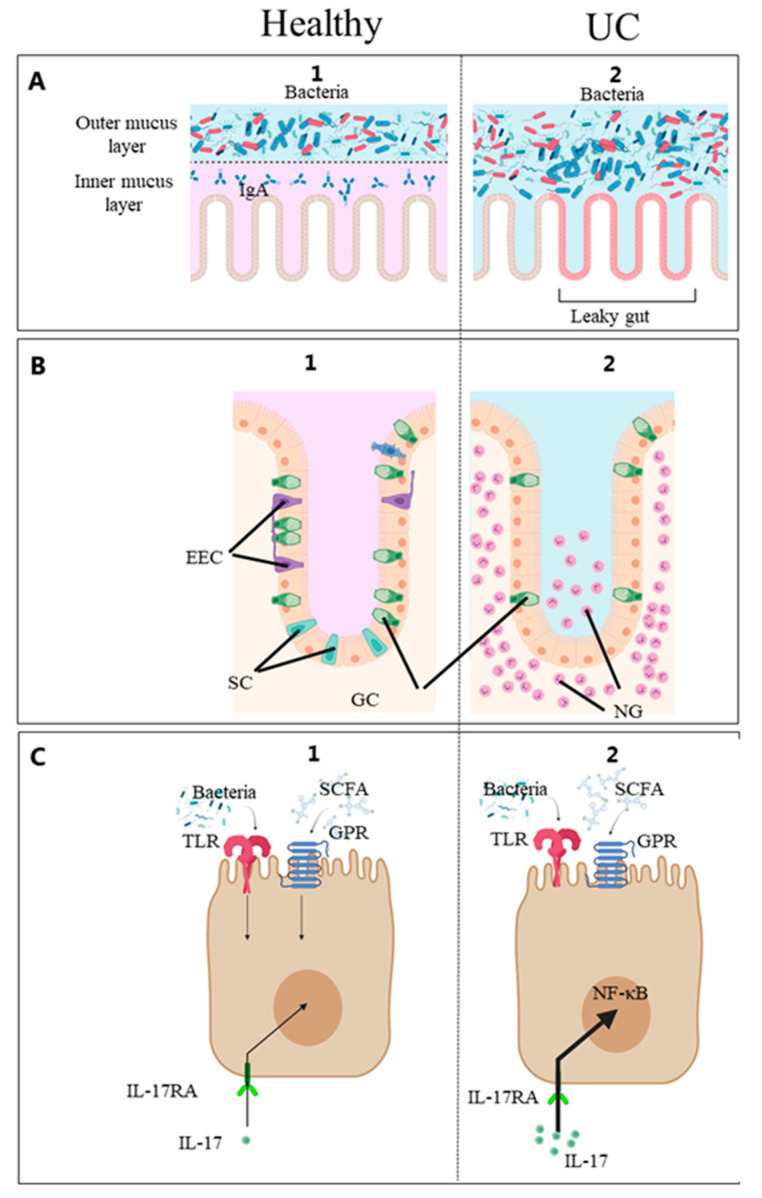
(**A**) Mucusa, (**B**) Epithelium, (**C**) Epithelial cell. Diet, gut microbes and patient factors interact at the mucosal surface. Schematic diagram of the intestinal mucosa constituting the intestinal barrier and immune system [1]. From the luminal side, it consists of the mucus and epithelial lining overlying the connective tissue. 1. (healthy) and 2. (UC). The outermost layer from the lumen side is the mucus. In the healthy gut, commensal microorganisms interact with the outer mucus layer and do not reach the inner mucus layer or epithelial cells. In IBD, the number of GCs is reduced, and this barrier is compromised, giving rise to the condition commonly described as a “leaky gut”. Certain microbial molecules activate the Toll-like receptors (TLR), and certain metabolites such as short-chain fatty acids (SCFA) activate the G protein-coupled receptors (GPR) on the intestinal epithelial cells. Enteroendocrine cells (EEC) monitor the gut microbiota and regulate inflammatory processes [27]. These processes stimulate the innate immune system resulting in gut inflammation by the pro-inflammatory IL-17 stimulating the IL-17 receptor A (IL-17RA), and neutrophilic granulocytes (NG) accumulate in the intestinal mucosa. EEC, enteroendocrine cells; GC, goblet cells; IL-17, interleukin-17; IL-17-RA, IL-17 receptor A; NF-ĸβ, nuclear-factor kappa beta; NG, neutrophilic granulocytes; GPR, G protein-coupled receptors; SCFA, short-chain fatty acids; TLR, Toll-like receptors; UC, ulcerative colitis. Created with Biorender.com (accessed on 2 May 2023).

**Table 1 ijms-24-11217-t001:** Dietary factors involved in patient–diet–microbial interactions.

	Function	Refs.
**Dietary**		
Ultra-processed foods, emulsifiers,	Associated with IBD	[28]
Western diet	Associated with IBD, flares	[29,30]
Mediterranean diet	Improve inflammation	[31]
Fibre intake	Feed anti-inflammatory bacteria	[31,32]

**Table 2 ijms-24-11217-t002:** Suggested biomarkers involved in patient–diet–microbial interactions.

	Function	Refs.
Microbiome		
SCFA producers, proinflammatory bacteria and fungi	Predict treatment response	[33,34]
Microbial composition and metabolites	Treatment response or progression	[35]
Faecal species richness, Candida or Caudovirales abundance, donor microbial profile similarity or biotin (vitamin B7)	FMT treatment response	[36]
Reduction in alpha diversity, abundance of Firmicutes	Predict postoperative recurrence in CD	[37]
Faecalibacterium and Bacteroides enrichment	Predict treatment response	[23]
Various specific bacteria	IBD diagnosis and prognosis	[38]
Microbial richness	Predict treatment response	[39]
Microbiome risk profiles	Predict treatment response	[24]
*Klepsiella pneumonia*	Associated to IBD	[40]
AIEC	Associated to IBD	[41]
Bacterial and Fungal Profiles	Predict treatment response	[33,39]
Metabolic profiles of bile acids, lipids and SCFAs	Predict treatment response	[25]
**Patient factors**		
**Genetic**		
Variants in *TNFSF4/18*, *PLIN2*, *NOD2*, *ATG16L1*, *TLRs* and *IL23R*	Predict treatment response	[42]
*IL-1B, IL-6, IFN-gamma, TNFRSF1A, NLRP3, IL1RN, IL-18, JAK2, LR2, TLR4, NFKBI*	Predict treatment response	[43]
*NOD2*, *CARD9* and *RIPK2*	Microbial sensing	[23]
*C1orf106* and *HNF4A*	Intestinal barrier function	[23]
Variants in *PIGR*, *NFKBIZ*, *IL17RA* and *TRAF3IP2*	Predict IBD-associated colon cancer	[44,45,46]
**Epigenetics**		
m6A modification	Predict prognosis	[47]
**RNA metabolism**		
HP1γ	Predict treatment response	[48]
**Immunologic**		
Faecal and serum calprotectin	Discriminate between the inflammatory and noninflammatory gut; track disease activity; treatment response	[43,49]
Anti-microbial antibodies	Predict disease development	[50]
Blood calprotectin, S100A12	Diagnosis and disease maintenance	[51]
Blood and faecal microRNAs	Predict treatment response	[43]
Redox biomarkers	Whole-body redox status	[52]
**Mucosal**		
TNF-α, IL-17A, IL-17R, OSM, OSMR, TREM1	Predict treatment response	[43]
Gasdermins	Intestinal barrier function	[53,54]
Mucosal FoxP3	Predict treatment response	[43]
NETs	Track disease activity	[55,56,57]
MPO, lactoferrin	Track disease activity	[58,59]
Mucosal F. prausnitzii	Predict treatment response	[43]
MCPIP1	Increase intestinal inflammation	[60]
**Urine**		
LMR	Predict disease development	[61]

Abbreviations; adherent-invasive *Escherichia coli*, AIEC; Faecal microbiota transplantation, FMT; heterochromatin Protein 1γ, HP1γ; interleukin, IL; MCPIP1, Monocyte chemotactic protein-1-induced protein 1; myeloperoxidase, MPO; N6-methyladenosine, m6A; neutrophil extracellular traps, NETs; Oncostatin M, OSM; OSMR, Oncostatin M Receptor; tumour necrosis factor, TNF; urinary fractional excretion of lactulose-to-mannitol ratio, LMR; short-chain fatty acids, SCFA; Triggering receptor expressed on myeloid cells 1, TREMI.

## Data Availability

No datasets were generated or analysed as part of this study.

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
