# Peer review of "Investigating the Crime Scene—Molecular Signatures in Inflammatory Bowel Disease"

_ijms, 2023, doi:10.3390/ijms241311217_

Round 1
Reviewer 1 Report
Dear Authors,
congratulations on your work. It is a good example
of a comprehensive review in this growing health issue.
In the section regarding biomarkers, instead just citing references, it would be more appropriate to expand and further elaborate these biomarkers.
Author Response
Thank you for pointing this out. We have now elaborated on these biomarkers and added (section 4.5): For example, leucine-rich alpha-2 glycoprotein can assess endoscopic activity in CD patients and is a reliable marker of endoscopic remission. Another study suggested that stool Chymotrypsin C, Gelsolin and Rho GDP-dissociation inhibitor 2 (RhoG-DI2) correlate with the levels of intestinal inflammation. The authors found that Gelsolin and RhoGDI2 in CD, and RhoG in UC, had higher sensitivity and specificity than faecal calprotectin in discriminating between patients and controls
Reviewer 2 Report
In this review, Andersen et al discuss the current status of research on Inflammatory Bowel Disease (IBD)- introduction, factors involved in disease onset and progression, biomarkers and interactions between diet, genetics, gut microbiome and immune system. They also discuss the cross talk between the different ileal cells, like Paneth cells and Goblet cells, and these factors and how that can promote IBD. The review is well-researched and well-written and very easy to follow through. They review both the known and the unknown (the gap in understanding) very well using tables and figures.
Author Response
Thank you so much for your kind words.
Reviewer 3 Report
Thank you for submitting your review manuscript to International Journal of Molecular Sciences. As the authors mentioned, many factors affect Inflammatory bowel diseases (IBD). Here, they dealt with these factors and discussed the strategy for disease management. It is well-organized and good to publish in IJMS if some parts are edited more.
- There are some typos (e.g., line 65, diat)
- It would be good to explain why the prevalence is increasing in Asia. That can elaborate on why these factors are related with the IBD.
- You might want to edit the box from line 61 for reading better.
- Line 119-122: Sounds like the repetition of the previous part.
- Line 121: I think the authors might want to give an example of biomarkers.
- Line 144-145: It is the definition of biomarkers.
- Table 1. Dietary factors and biomarkers should be presented in separated tables.
- In 3.1. Diet and IBD, it would be better to explain why these types of diets can influence IBD.
- In 3.2. The authors might want to make a table for the bacteria, fungi, and virus which affect IBD.
- As far as I know, the immune system greatly influences IBD. The 3.3 looks too weak and short to deal with the importance of it. The authors should elaborate on it more.
- I like part 4 for demonstrating the importance of interaction and patients factors. But I think combining these factors (line 419) sounds too vague. It would be nicer to suggest the concrete strategies for the better IBD management.
A minor editing of English is needed.
Author Response
Thank you for your helpful comments which have helped us improve the manuscript.
Typos. We have been through the manuscript to correct typos.
Asia. Added: For example, future strategies for better management of IBD could include characterising diversity among patients by e.g. analysing specific genetic variants related to the mucosal barrier and microbial handling together with microbial risk scores indicative of inflammation. In addition, the changing patterns of IBD across the world such as the rising prevalence of IBD in Asia can provide insights into IBD causes.
Line 119-122 and Line 121: Thank you for noticing. It has been replaced with: For example, microbiome risk profiles may have the potential to identify similar groups of patients.
Line 144-145: Rephrased This review argues that biomarkers reflecting these interactions (lifestyle, gut mi-crobes, and the gut barrier immune system) can serve as diagnostic, prognostic, pre-dictive or monitoring tools to enhance the management of IBD
Table 1 has been divided into two tables.
In 3.1. Added: A wholegrain fibre diet increased the level of butyrate in overweight individuals.
In 3.2. We considered added potential fungi and virus to the existing supplemental table of bacteria affecting IBD. However, we decided not to do it as the information on the specific effects of fungi and virus in IBD is still sparse.
3.3. Added: In recent years, new epithelial cell types such as intercrypt goblet cells, microfold-like (M-like) cells, BEST4+ cells, and Tuft cells have been identified. Tuft cells appear to be critical for specific immunologic responses. M-like cells are rarely found in healthy colons but are reported to be expanded 17-fold in inflamed colons. BEST4+ cells were identified as a new population of human intestinal epithelial cells by single-cell RNA-seq technology. Histologic analysis revealed their localization in the crypt top. The functional role of BEST4+ cells remains unknown, but they may be associated with bicarbonate export and pH-sensing function based on their gene expression. Finally, the gut epithelial basement membrane is a specialized matrix that supports and separates the epithelial cells from the interstitial space and is also considered important in maintaining the epithelial barrier.
Part 4 (l 419). Added (to 5): For example, future strategies for better management of IBD should include characterising diversity among patients by e.g. analysing specific genetic variants related to the mucosal barrier and microbial handling together with microbial risk scores indicative of inflammation. In addition, the changing patterns of IBD across the world such as the rising prevalence of IBD in Asia can provide insights into IBD causes.